# Colloidal gel elasticity arises from the packing of locally glassy clusters

Kathryn A. Whitaker [1,5], Zsigmond Varga [2], Lilian C. Hsiao[3], Michael J. Solomon[4], James W. Swan[2] & Eric M. Furst [1]

Colloidal gels formed by arrested phase separation are found widely in agriculture, biotechnology, and advanced manufacturing; yet, the emergence of elasticity and the nature of the arrested state in these abundant materials remains unresolved. Here, the quantitative agreement between integrated experimental, computational, and graph theoretic approaches are used to understand the arrested state and the origins of the gel elastic response. The micro-structural source of elasticity is identified by the *l*-balanced graph partition of the gels into minimally interconnected clusters that act as rigid, load bearing units. The number density of cluster-cluster connections grows with increasing attraction, and explains the emergence of elasticity in the network through the classic Cauchy-Born theory. Clusters are amorphous and iso-static. The internal cluster concentration maps onto the known attractive glass line of sticky colloids at low attraction strengths and extends it to higher strengths and lower particle volume fractions.

[1] Department of Chemical and Biomolecular Engineering, University of Delaware, Newark, DE 19716, USA. [2] Department of Chemical Engineering, Massachusetts Institute of Technology, Cambridge, MA 02139, USA. [3] Department of Chemical and Biomolecular Engineering, North Carolina State University, Engineering Building I, Raleigh, NC 27695, USA. [4] Department of Chemical Engineering, University of Michigan, Ann Arbor, MI 48109, USA. [5] Present address: Dow, 1702 Building, Midland, MI 48667, USA. Correspondence and requests for materials should be addressed to J.W.S. (email: jswan@mit.edu) or to E.M.F. (email: furst@udel.edu)

In soft solids composed of colloidal suspensions, emulsions, foams, and biomaterials, elasticity is governed by the spatial distribution and interactions among amorphous mesoscale structures[1]. Identifying and understanding the behavior of these fundamental building blocks are the underlying challenges for developing structure-property relations that are essential to controlling and tailoring the mechanics of such materials. Among soft solids, an important and ubiquitous class are colloidal gels, in which attractive interactions between suspended colloidal particles drive a thermodynamic instability that promotes aggregation, arresting in a space spanning network structure possessing unique mechanical and transport properties[2,3]. Commonly, phase separation is induced by the addition of non-adsorbing polymer to a suspension of repulsive colloids by the well-known depletion interaction[4–7]. Depletion gels are often found in industrial processes and products where fine solids are dispersed in polymer solutions, including agrochemicals, consumer care products, and pharmaceuticals, and have frequently served as model experimental systems[8,9].

In applications, the rheology of a gel is its principal material property of interest, including its elasticity[10] and yielding[11]. At low volume fractions and strong interaction energies between particles, colloidal gels are effectively modeled as fractal flocs formed through diffusion-controlled aggregation processes, which grow together to form a percolating microstructure[12]. The flocs are the principal load bearing units of the gel and theories connecting the floc architecture to the gel modulus remain a state-of-the-art description[13,14]. Yet, there exists no definitive micro-structural theory for the elasticity of colloidal gels at higher volume fractions and lower strengths of interaction. What are the fundamental structural units imparting elasticity to the network, and what physical principles govern their formation?

Linear elasticity in depletion gels has been postulated to result from the spatial organization of particles into locally dense clusters, each cluster acting as a rigid, mechanical unit that propagates the elastic deformation[10,15–17]. For instance, the work of Zaccone, Wu, and Del Gado[16] theorized that the role of cluster-cluster contacts were central to gel elasticity. The authors showed that one model of cluster microstructure, based on contact number distributions for hard spheres and the Cauchy-Born theory for the affine elastic response of amorphous solids[18], can fit experimental measurements of the shear modulus. Similar models with different approaches to enumerating clusters and cluster contacts based on mode coupling theory (MCT) were also explored by Ramakrishnan and co-workers[10]. Signatures of clustering are evident in light scattering[10] and confocal microscopy[19], which probe long-range fluctuations in colloid number density, and active microrheology, which examines the elastic deformation in response to a local perturbation[20].

In the present work, we combine experimental, computational, and graph-theoretic approaches to systematically identify clusters in depletion gels and show that they indeed constitute the fundamental structural units that lead to the gel's elastic response. For the same inter-particle interactions, the structure and elastic modulus of the experimental and simulated gels agree quantitatively. Our experiments and simulations show that the gel elastic modulus is a convex function of the inter-particle attraction. However, this property cannot be accounted for by any macroscopic structural change or the change in inter-particle bond stiffness. To resolve this important puzzle, we use a graph theoretic approach to identify the fundamental elastic units of the gels, a set of clusters, and to measure the density of cluster-cluster contacts. The physical size of the clusters is independent of the strength of the depletion attraction, but their number density and the number of cluster-cluster contacts grows with increased attractive strength. These changes, when combined with the

Cauchy-Born theory, yield a prediction of the elastic modulus with the correct convexity and in quantitative agreement with experimental measurements and calculations from simulations. Additionally, we find that the internal cluster structure becomes less dense with increasing attraction. The volume fraction of colloids within the clusters falls on and then extends the attractive glass line (AGL) of colloids determined via MCT, suggesting that the arrest of the gel forming process is due to glassy physics within the fundamental elastic units. Colloidal gels formed via arrested phase separation should be viewed as dense packings of locally glassy grains. There is a striking similarity to polycrystalline solids: elasticity and, potentially, plasticity emerges from the few weak bonds between clusters[21].

## Results

**Gel structure, rheology, and interparticle interactions**. The structure, rheology, and interparticle interactions for the depletion gels are summarized in Fig. 1. In experiments, a recently developed model depletion gel is employed, which enables the rheology, microstructure, and particle interactions to be measured in concert for complete determination of the microscopic properties and macroscopic elastic response[22]. In simulations, a high-performance Brownian dynamics simulation algorithm generates representative depletion gel structures and enables us to compute the elastic modulus of those gels[23,24].

Colloidal gels formed by arrested phase separation (Fig. 1a, experiment; 1b, simulation) exhibit no significant change in number density correlation length and only subtle changes in local bonding structure with increasing strength of attraction between the colloids. However, the elastic modulus of the gels measured experimentally and computed in simulations over the same range of attractions varies by more than a factor of 5 (Fig. 1c). Figure 1d depicts the number density fluctuations within spherical volumes of radius $r$ measured from particle tracking applied to confocal microscopy of depletion gels and computed from results of Brownian Dynamics simulations. For large $r$, all density fluctuations decay as the power-law $r^{-3}$, reflecting a homogeneous structure on large length scales. Below the correlation length $\xi$ (discussed below) shown in Fig. 1e, the number density fluctuations scale with a different power-law sensitive to the local structure of the gels. In both simulations and experiments, these differences are plainly evident. Over length scales similar to the particle size, the experimentally measured variations in the number density fluctuations are overwhelmed by statistical noise in the particle tracking algorithm arising from static and dynamic tracking errors.

The correlation length $\xi$ (Fig. 1e) is visible in the experimental micrographs and renderings of the simulations. We calculate it from the peak of the static structure factor computed from both the experiments and the simulations. Across a broad range of interaction strengths, the micro-structural correlation length is nearly constant and in close agreement between experiment and simulation. The value $\xi \approx 10a$, where $a = 0.560\ \mu m$ is the colloid radius, is commonly observed in experiments with gels formed via arrested phase separation[10]. It is well understood that a diffusion-limited aggregation process precedes the arrest of the gel in its terminal, elastic state, which freezes in the same correlation length regardless of the strength of the attractive interaction[9,25,26]. But the key question remains: How are the local number density fluctuations to be understood, and what impact do these have on the elasticity of the colloidal gels?

**Cauchy-Born theory**. In order to understand the emergence of elasticity in the particle network, a model of gel elasticity can be formulated from the Cauchy-Born theory, in which

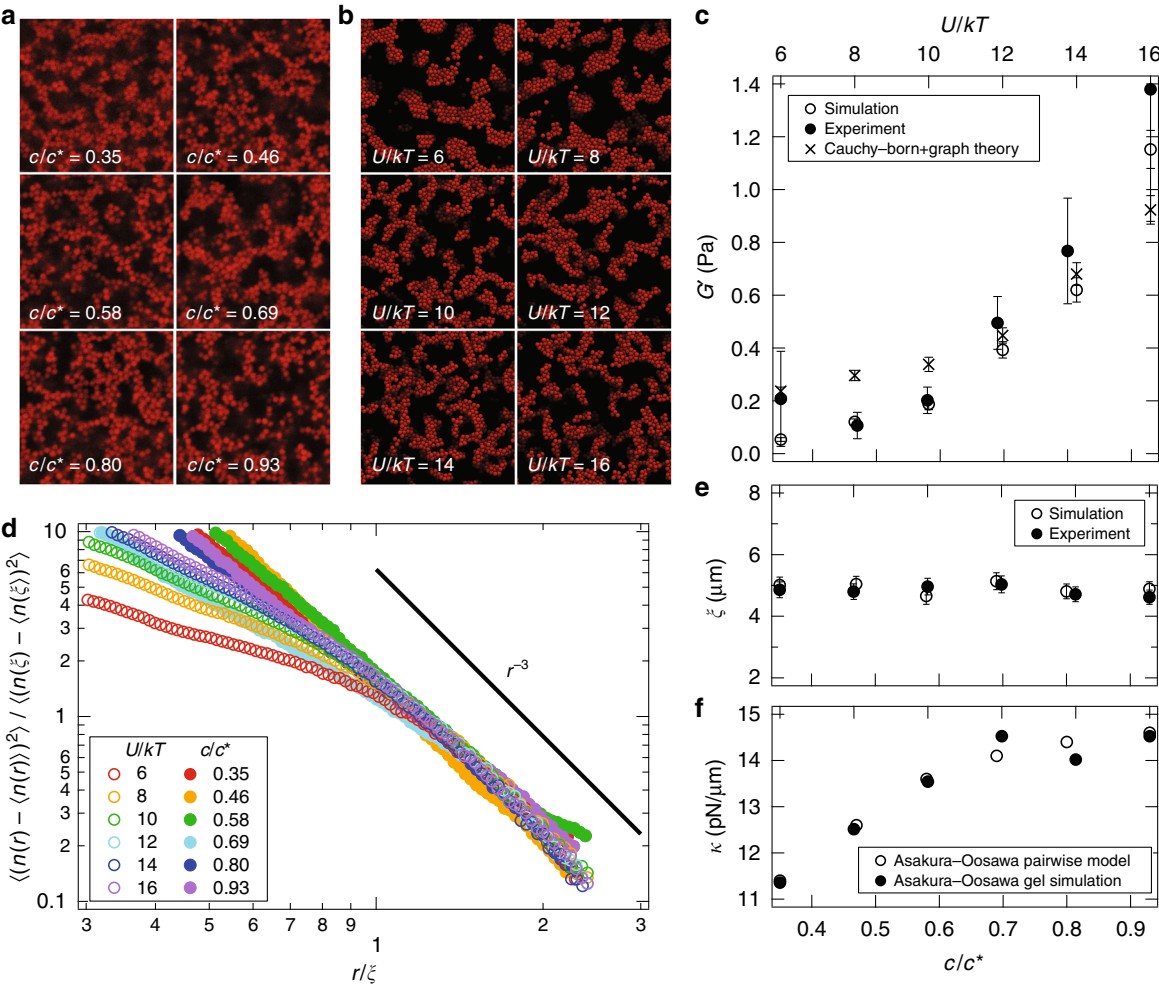

**Fig. 1** The microstructure and rheology of simulated and experimental gels. **a** Confocal micrographs and **b** simulation snapshots of depletion gels at different polymer concentration and fixed volume fraction $\phi = 0.20$. **c** The elastic modulus of depletion gels measured in experiment (solid) and computed via simulation (open) as a function of depletant concentration and strength of attraction. The predictions of the graph-decomposition and Cauchy-Born theory (crosses) compare well with the measurements. The error bars of the experiments are calculated from three independent measurements. Similarly, the error bars for the simulations are the 95% confidence intervals from data collected across multiple gels under the same conditions. The frequency dependence of the experimental and simulated shear moduli are discussed in Supplementary Note 1 and compared in Supplementary Fig. 1. **d** Variance in the number density of particles within a spherical boundary of radius $r$ measured from experiments (solid) and simulations (open). The solid line is a power-law decay for a geometrically dense structure. **e** The characteristic correlation length of number density fluctuations measured in experiments (solid) and computed via simulations (open). **f** The stiffness of inter-particle bonds computed from the Asakura-Oosawa theory and direct application of the equipartition theorem (open) or measured directly in simulations of arrested gels (solid)

the elastic modulus arises from a few weak connections between locally rigidified clusters[10,15,16]. The Cauchy-Born elastic modulus is written as the product of three physical quantities: $G' = 4n_e\kappa \langle r^2 \rangle$, where $n_e$ is the number density of elastically active bonds in the material, $\kappa$ is the bond stiffness, and $\langle r^2 \rangle = (1/3)\int(r_x r_y + r_y r_z + r_x r_z)P(\mathbf{r})d\mathbf{r}$ is a squared length scale associated with the elastically connected domains[18]. Formally, it is an average over the cross-second moments of the vector $\mathbf{r}$, connecting each cluster's geometric center to the point at which it is bonded to its neighboring clusters taken over the distribution of cluster-cluster contacts, $P(\mathbf{r})$.

It is natural to expect $\langle r^2 \rangle$ to scale with one of the two length scales evident in a depletion gel: the particle size, $a$, or the correlation length, $\xi$. Since neither varies significantly with inter-particle interaction, changes in length scale cannot explain the variation of the elastic modulus in gels formed by arrested phase separation. The bond stiffness $\kappa$, can be estimated from the inter-particle potential through equipartition by computing $kT/\langle h^2 \rangle$,

where $kT$ is the thermal energy and $\langle h^2 \rangle$ is the average of the squared surface to surface separation of a pair of particles whose inter-particle distance satisfies a Maxwell-Boltzmann distribution. It may also be inferred in the simulations of arrested gels through explicit calculation of $kT/\langle h^2 \rangle$. Both measures are plotted in Fig. 1f. The bond stiffness with respect to the strength of inter-particle interaction increases modestly and has the opposite concavity of the elastic modulus, and thus it alone cannot explain how elasticity in the gels develops. What remains from the Cauchy-Born theory to explain the change in elasticity is a change in the number density of elastically active bonds.

Visual inspection of gel structures measured experimentally and computed in simulations offers no clear delineation of cluster and inter-cluster bonds beyond the correlation length $\xi$ associated with the number density fluctuations. Laser tweezer experiments have identified rigidly clustered regions with similar characteristic length scale through mechanical interrogation of the network by an oscillatory driving force[20]. This form of mechanical

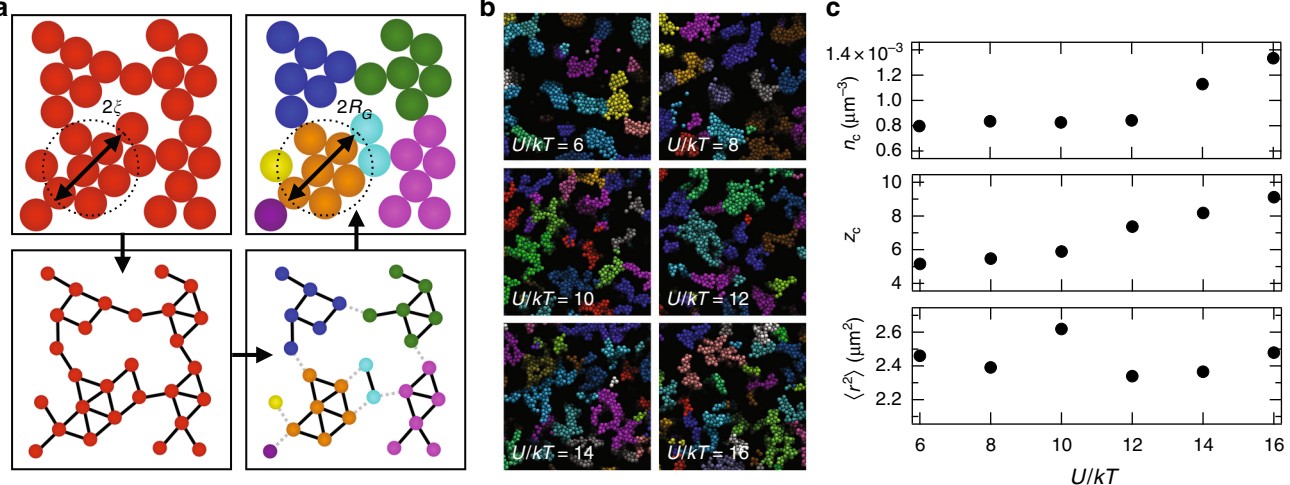

**Fig. 2** Graph decomposition of depletion gels formed by arrested phase separation. **a** Illustration of the $l$-balanced graph partition algorithm applied to a model gel. In counter-clock-wise order: the network is converted into a graph with particles representing vertices and bonds representing edges, the graph is partitioned into subgraphs containing no more than $l$ vertices ($l = 6$ here), and the subgraphs are used to color the depletion gel. **b** Snapshots from simulation data at varying $U/kT$ of depletion gels after coloring with the graph partitioning scheme. The subgraph size $l$ is chosen so that the correlation length $\xi$ equals the average radius of gyration of the clusters found via partitioning. **c** The subgraph number density, the average number of contacts between subgraphs, and the averaged square subgraph contact length: $\langle r^2 \rangle$

interrogation is purely local and identifies only single clusters, and cannot, in a practical way, provide a statistical description of the cluster number density or their inter-connectedness. Yet, these two quantities: the number density of rigid clusters, $n_c$, and the average number of cluster-cluster contacts per cluster, $z_c$, are sufficient to determine the remaining parameter for the Cauchy-Born theory: the number density of elastically active bonds, $n_e = \frac{1}{2} n_c z_c$. An alternative means of identifying rigid clusters and their connectedness is needed. In the present work, we employ a graph theoretic approach[27,28].

**$l$-balanced graph partition of gels**. To calculate $n_c$, $z_c$, and $\langle r^2 \rangle$, we apply the $l$-balanced graph partition (Fig. 2a) to represent the rigidly connected clusters comprising the gel in a form that encodes the topology of the inter-particle bond network but that is abstracted from the particles' physical spatial arrangement. An unweighted, undirected graph representing each simulated gel is built by associating every particle with a vertex and every bond between two particles with an edge connecting the corresponding vertices. Decomposition proceeds by cutting edges to divide the graph into clusters containing no more than $l$ particles. The partition is chosen such that there are nearly equal numbers of vertices in each cluster (balanced) while cutting the fewest possible edges. Subgraphs given by the partition represent the most loosely connected set of particle clusters containing $l$ particles. The bound of $l$ can be chosen arbitrarily, and here we use bisection to efficiently find a value $l^*$ such that the average radius of gyration, $R_G$, of the clusters identified in the subgraphs is equal to the correlation length of the gel, $\xi$. The number density of clusters is simply $n_c = n/l^*$, (where $n$ is the number density of particles) and is plotted in Fig. 2c. Importantly, the decomposition shows that the number of rigid clusters, and thus $n_c$, is an increasing function of $U/kT$. With the cluster size and overall number density fixed, the internal density of particles internal to the clusters is a decreasing function of $U/kT$. The colored clusters identified by the graph partitioning (Fig. 2b) exhibit this property.

The connections between subgraphs determines the average cluster contact number $z_c$ and the square length scale $\langle r^2 \rangle$. Figure 2c depicts $n_c$, $z_c$, and $\langle r^2 \rangle$ computed from the graph

decomposition. Like the correlation length $\xi$, $\langle r^2 \rangle$ is not a systematic function of the attraction strength. However, $z_c$ is an increasing function, and crucially, the product $n_c z_c$ is a convex function of $U/kT$, such that the number of elastically active contacts alone is able to reproduce trends in the elastic modulus measured from experiments and simulations. When these measures of the connectivity among the fundamental elastic subunits are substituted into the Cauchy-Born theory (Fig. 1c), the elastic modulus is reproduced quantitatively as well.

We compute a histogram of the number of bonds formed by particles within the individual clusters to verify that the clusters identified by the $l$-balanced graph decomposition are themselves rigid (Fig. 3). This distribution reveals that the mean number of bonds per particle exceeds six for all cases studied. For particles with central interactions, six bonds are necessary for iso-staticity and thus rigidity. The contact number distribution cannot prove definitively that the clusters identified by the decomposition are internally rigid. However, that the mean number of contacts per particle in a cluster is larger than six is consistent with clusters having a rigid core. Specific notions of iso-staticity and rigidity of the whole gel structure or parts has been highlighted recently through alternative methods[29–31].

It is important to recognize that the $l$-balanced graph decomposition is not unique. For example, consider the graph with five vertices connected in a ring. The $l$-balanced graph decomposition with subgraphs having three or fewer vertices ($l = 3$) could result in any pair of subgraphs containing three and two vertices selected from the ring by cutting just two edges. We employ a spectral decomposition method that determines the decomposition through $k$-means clustering using a stochastically sampled initial condition. Degeneracy of the decomposition should only affect the loose connections between rigid subunits and not the subunits themselves. The effect of degeneracy in the decomposition on the predicted macroscopic elasticity was quantified by determining 95% confidence intervals on the inferred quantities from five replicates of the decomposition process. These uncertainties are smaller than the size of the symbols in Fig. 2c, but are propagated forward to the elastic modulus determined from the Cauchy-Born theory in Fig. 1e. During the process of bisection for determining the value of $l$, we

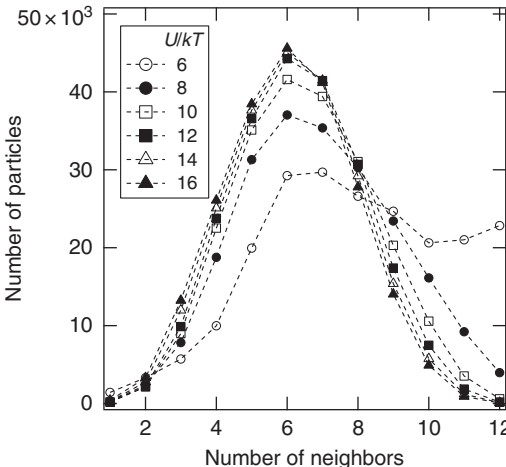

**Fig. 3** The nearest neighbor distribution for particles within clusters. The $l$-balanced graph decomposition identifies minimally connected clusters among the different gels studied. The analysis itself is agnostic to whether the clusters themselves are rigid. By identifying minimally connected clusters in a homogeneous network, there is a bias toward clusters with large number of internal bonds. To quantify this effect and confirm that the clusters are indeed rigid, we examine each cluster in isolation and compute the number of bonded neighbors each particle within the cluster possesses. This number excludes any neighbors that might reside within another cluster. The plot shows this distribution for all the simulated gels. Clearly, the mean number of nearest neighbors is larger than six for all cases. Therefore, the clusters exceed the Maxwell iso-staticity condition and contain a rigid core that can serve as an elastic sub-unit for the Cauchy-Born theory

observe that the mean radius of gyration of clusters varies monotonically in the neighborhood of $l^*$, which itself is much smaller than the total number of particles in the gels. Therefore, based on smoothness of the objective function and reproducibility of the decomposition, we conclude that the graph decomposition is stable and insensitive to the precise value of $l^*$ when the gel contains a large number of clusters.

While it is ultimately desirable to use the experimental confocal image data to perform an identical cluster analysis, the present experimental system sacrifices some particle tracking resolution (index mismatch) to enable both rheology and optical trapping for direct measurements of the particle interactions by laser tweezers[22]. This increases both the static and dynamic contributions to the variation in particle positions. Figure 1d clearly shows the small wave vectors and short length scales of these contributions, while both experiment and theory capture the long-range structure of the gel.

## Discussion

The model data in Fig. 1c are the quantity: $2n_c z_c \kappa \langle r^2 \rangle$, with $n_c z_c$ and $\langle r^2 \rangle$ determined via graph theoretic methods. The results show that Cauchy-Born theory yields a prediction of the elastic modulus with the correct convexity and in quantitative agreement with experimental measurements and calculations from simulations. The graph decomposition, which is agnostic to the absolute spatial arrangement of the colloids, has revealed the underlying elastic structure of these heterogeneous gels. Similar graph theoretic approaches have proven useful for understanding the mechanical properties of other amorphous solids such as granular packings[32].

The graph theoretic approach provides further insight into the microstructure of gels formed during arrested phase separation on length scales below the correlation length, $\xi$. The number density of clusters $n_c$ can be used to define an effective cluster volume fraction $\phi_c = 4\pi n_c \xi^3/3$, representing the fraction volume in the gel occupied by the elastic subunits. Through conservation of particle number, the volume fraction of colloids internal to the subunits is $\phi_g = \phi/\phi_c$. Because $n_c$ is an increasing function of the strength of inter-particle attraction, the internal volume fraction of the clusters is a decreasing function of the same. That is, the elastic subunits contain fewer particles with increasing strength of attraction. Figure 4a depicts both $\phi_c$ and $\phi_g$ determined from the graph decomposition. Following the work of Zaccone, Wu, and Del Gado[16], an approximation for $\phi_c$ is also calculated by solving the implicit equation $G' = \phi_c z_c(\phi_c)\kappa/(5\pi\xi)$, at each strength of attraction and measured value of $G'$. This equation is a re-stated form of the Cauchy-Born theory with $G'$ the modulus determined in the simulations, $\langle r^2 \rangle = \xi^2/15$, and $z_c(\phi_c)$ drawn from the hard-sphere equation of state[33,34], in which the radial distribution function at contact, $g(2;\phi) = \frac{1-\frac{1}{2}\phi}{(1-\phi)^3}$, gives $z_c(\phi) = \phi g(2;\phi)$. The resultant volume fraction of clusters and their internal volume fraction are also plotted in Fig. 4a. The hard-sphere approximation provides a reasonable estimate of the volume fraction of elastic subunits and their internal volume fraction; both differ from the graph-theory result by roughly 10%. The data in Fig. 4a come from the simulated $G'$ to provide a direct comparison between different models of the cluster architecture: graph decomposition and hard sphere equation of state. A quantitatively equivalent set of cluster volume fractions can be generated using the hard-sphere approximation and experimentally measured values of $G'$, which are indistinguishable from the the simulated values within the experimental uncertainty.

Remarkably, the internal volume fraction of clusters $\phi_g$ extrapolate from the AGL calculated from MCT, which includes a bilinear coordinate transformation to superimpose the MCT theory predictions onto Monte Carlo simulations[35,36]. It is important to note that glassiness is conventionally defined in terms of long or diverging relaxation times and not merely the density of a phase. We observe in both experiments and simulations that the gel as a whole is arrested. This highlights the importance of work that tracks the morphology and dynamics of individual clusters over longer time scales than studied here, in order to understand how clusters form, relax, and eventually arrest[30]. In Fig. 4b, each interparticle attraction strength corresponding to a depletant concentration is represented by the reduced Baxter temperature $\tau_B$[3,9,37–39]. The Baxter temperature is related to the reduced second virial coefficient $B_2^*$ by $\tau_B = \frac{1}{4(1-B_2^*)}$, where $B_2^* = \frac{B_2}{\frac{2}{3}\pi\sigma_{eff}^3}$, $\sigma_{eff}^3$ is the effective hard sphere diameter, and $B_2$ is the second virial coefficient calculated from the pair interaction potential $U(r)$ by $B_2 = 2\pi \int_0^\infty r^2 [1 - e^{-U(r)/kT}] dr$. The Asakura-Oosawa potential is used to calculate $B_2^*$ for the experimental system of PMMA in CH and CHB (mass fraction $w_{CH} = 0.37$) with polystyrene depletant. Based on the principle of corresponding states for short-range attractive interactions[40], the particle volume fraction and reduced Baxter temperature $\tau_B$ for each experiment is plotted on the phase diagram for adhesive hard spheres (AHS) with open black circles in Fig. 4b. The solid black circles represent the corresponding $\phi_g$ values derived from the cluster model. We also plot $\tau_B$, $\phi$, and $\phi_g$ obtained from fitting the cluster model to the elastic moduli of depletion gels from octadecyl silica nanoparticles in decalin[10] (Details of the model fits are provided in Supplementary Note 2). The results are remarkably consistent between depletion gels that differ substantially in size and chemistry, but that otherwise correspond to

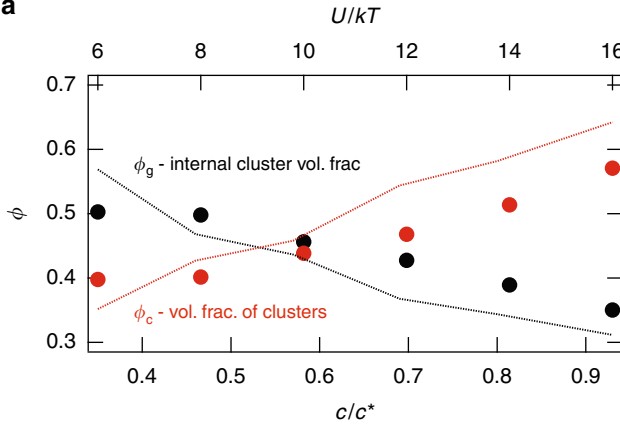

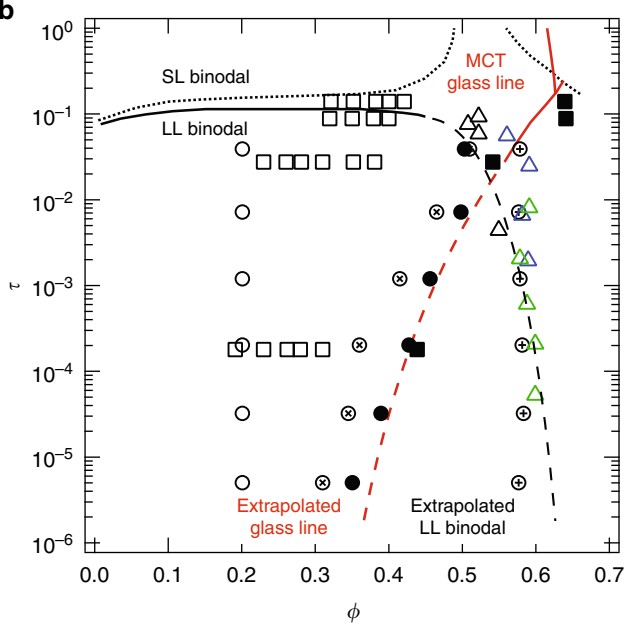

**Fig. 4** The internal cluster particle concentration. **a** The cluster densities $\phi_g$ (black) and volume fractions of clusters $\phi_c$ (red) determined from simulations and application of the graph theory. The lines are approximations of $\phi_g$ (black) and $\phi_c$ (red) found by substituting the hard sphere equation of state into the Cauchy-Born theory with $\langle r^2 \rangle = \xi^2/15$ and $G'$ given by the simulation results. Nearly identical values are found when using the experimentally determined modulus. **b** Initial volume fraction and reduced temperature of gel samples (open circles) and cluster densities $\phi_g$ determined in this work (closed circles, graph theory; circle-x, hard sphere equation of state). The square symbols reproduce the data of Ramakrishnan et al.[10] with open squares the experimentally measured particle volume fraction and closed squares the cluster densities inferred from application of the Cauchy-Born theory with the hard-sphere equation of state. The attractive and repulsive glass lines with a bilinear shift from the mode coupling theory (MCT) are drawn in solid red[35,36] while an extrapolation through the experimental data is indicated with the dashed red line. The AHS phase boundaries with the fluid-solid coexistence lines for $a/R_g = 10$ [6] are given by short dashed black lines. Fluid-fluid coexistence is indicated by the solid black line and its extrapolation by the dashed black line. The local volume fraction of particle-rich regions (shown as open circle-plus) is compared with values from ref. [9]: blue triangles are $\phi = 0.13$ and $\delta = 0.059$, green triangles are $\phi = 0.045$ and $\delta = 0.018$, and black triangles are $\phi = 0.045$ and $\delta = 0.059$, where $\delta$ is the range of the attraction relative to the particle radius

the same scaled range and strength of the interaction potential. Finally, without a graph-theoretic approach to identify gel clusters, we calculated the local packing fraction of dense particle strands in the network, following the approach of Lu et al.[9] These values, shown in Fig. 4b, are roughly constant at a volume fraction $\phi \sim 0.57$ and agree with their earlier study using PMMA colloids ($a = 0.560\,\mu\text{m}$) in density matched cyclohexyl bromide and decahydronaphthalene with tetrabutylammonium chloride.

The Cauchy-Born cluster model of gel rheology accurately describes the elastic modulus of depletion gels based on the bond rigidity, the cluster size, and the cluster density. As the attractive strength increases with depletant concentration, the gel modulus also increases. However, the higher modulus is due mainly to the lower density of particles within clusters, which coincides with the AGL extrapolated into the coexistence region of the phase diagram. These results are supported by a long line of evidence that there is an intimate connection between the gelation of colloids with short-range attraction and phase separation. For particles with centro-symmetric forces, a thermodynamic driving force in the two-phase region drives aggregation, and locally arrested density fluctuations, clusters that are sparsely connected, give the network elastic properties.

This work raises a number of interesting possibilities for engineering the mechanical strength of colloidal gels, similar to controlling crystallite grain size in metals[41,42]. During the quiescent formation of gels, the constant value of $\xi$ at different attractive strengths results from the diffusion-limited aggregation kinetics. By imposing a flow or external field, the cluster size can be altered and clusters can even be made anisotropic. Thus, one could rely on processing, with a focus on altering the cluster size $\xi$ or the cluster shape as a means of controlling the material stiffness. The Cauchy-Born theory has a more primitive tensorial form that can account for such anisotropies. The $l$-balanced graph decomposition provides the necessary tool to detect anisotropic clusters from which these anisotropies derive. Finally, the graph theory methods introduced here should also be powerful tools for identifying and characterizing meso-scale structures from real-space position data that dictate the mechanics of other networked colloidal materials, like capillary suspensions[43] and particle networks at interfaces[44–46].

## Methods

**Depletion gels.** Experiments are performed using a recently-developed model colloidal gel that enables the simultaneous measurement of the structure, rheology, and particle interactions[22]. The depletion gels consist of poly(methylmethacrylate) latex (PMMA) particles dispersed in a mixture of cyclohexane and hexadecane. With this model system, we measured the bulk shear modulus with a rotational rheometer, the microstructure using confocal microscopy, and the particle interactions with laser tweezers.

Depletion gels were prepared with PMMA particles ($2a = 1.11\,\mu\text{m} \pm 3\%$) suspended in a mixture of cyclohexane (CH) and cyclohexyl bromide (CHB) with the CH mass fraction $w_{CH} = 0.37$ at a particle volume fraction $\phi = 0.2$. Polystyrene ($M_w = 900,000$ g/mol, $c^* = 10.8$ mg/mL, $R_g = 32 \pm 2$ nm) is added at different concentrations ($c/c^* = 0.35, 0.46, 0.58, 0.69, 0.80$, and 0.93) to adjust the strength of the attraction between the particles. The results from the model system are compared with depletion gels reported by Ramakrishnan et al.[10] (Supplementary Note 2 contains information on the experimental system). The material properties of the two model systems are summarized in Table 1.

**Confocal imaging of gel structure.** An inverted confocal microscope (Nikon A1Rsi) equipped with a resonant scanner head and a high-speed piezo stage is used to image the colloidal gels. Gel samples are loaded into glass capillaries with 300 $\mu$m spacers to match the gap used in the rheometer. The top and bottom of the capillaries are glass coverslips suitable for microscopy (0.17 mm thickness). After loading, the gels sit quiescently for 30 min before imaging. 3D image stacks are taken from the bottom coverslip at a speed of 15 slices per second. The image stacks are 42 $\mu$m × 42 $\mu$m × 10 $\mu$m in dimension. The voxel dimensions are 83 nm × 83 nm × 83 nm.

Confocal image volumes are analyzed using feature finding code based on the particle tracking algorithm of Crocker and Grier[47], which uses a Gaussian mask to filter out digital noise and that identifies centroids based on their intensity maxima.

### Table 1 Model depletion systems compared in this work

|  | This work | Ramakrishnan et al.[10] |
| --- | --- | --- |
| Particles | PMMA | Octadecyl silica |
| Solvent | CH/CHB | Decalin |
| Radius, $a$ (μm) | 0.555 | 0.045 |
| Depletant | PS | PS |
| $M_w$ (g/mol) | 900,000 | Not reported |
| $R_g$ (nm) | 32 | 3.5 |
| $c^*$ (mg/mL) | 10.8 | Not reported |

The particle tracking is modified from the original two-dimensional algorithm to locate the center of a particle within a three-dimensional voxel with sub-voxel resolution. Errors in centroid location are determined in the $xy$- and the $xz$-planes using immobilized samples. The error is ±34 nm in the $xy$-plane and ±67 nm in the $xz$-plane. Particle position data are used to calculate several measures of the gel structure, including the particle contact number distribution and the number density fluctuations. Particles are considered to be nearest neighbors if they are closer than the distance that defines the first minimum in the radial distribution function, $g(r)$.

**Bulk rheology.** Oscillatory rheometry is performed on a stress-controlled rheometer (AR-G2, TA Instruments) at $T = 25$ °C. Colloidal gels are loaded onto a Peltier plate and a stainless steel parallel plate geometry ($d = 6$ cm, gap $h = 300$ μm) is lowered to the gap distance while rotating at $\omega = 1$ rad/s to minimize the formation of bubbles at the sample interface. The geometry minimizes confinement effects and a solvent trap is used to prevent the loss of the volatile solvent. We employ a correction factor for the nonhomogeneous strain rate in the parallel plate geometry[48]. We verified that gap size effects are negligible for $h > 150$ μm, which is determined by measuring the strain-dependent linear elastic modulus for poly (ethylene oxide) standards (Molecular weight $M_w = 2 \times 10^6$ g/mol, 4 wt%) at $h =$ 50, 100, 150, 200, 300, 400, and 500 μm, and for colloidal gel samples at $h = 300$ and 500 μm. Slip does not occur; a smooth fixture and a fixture with a sand-blasted surface (parallel plate geometry $d = 6$ cm) give identical moduli to within experimental error.

After sample loading, gels are pre-sheared unidirectionally for 1 min (strain = 37,700, shear rate = 628.3 s$^{-1}$). Oscillatory strain sweep (fixed angular frequency, $\omega = 10$ rad/s) and frequency sweep measurements (fixed strain amplitude, $\gamma = 0.01$) are performed after waiting 30 min. The waiting time is chosen to allow direct comparison between the gel microstructure and the rheological measurements on the gels. The average of three independent measurements of the gel rheology are made for each depletant polymer concentration.

**Brownian dynamics simulations.** Computer simulations are performed using a recently developed a method for rapid calculation of hydrodynamic interactions (HI) in suspensions of mono-disperse spheres[24]. The positively-split (PSE) algorithm makes the cost of computing Brownian displacements in simulations of colloidal scale particles with HI comparable to the cost of computing deterministic displacements in freely draining simulations. Here, the Rotne-Prager-Yamakawa tensor (RPY)[49] is used to account for the long-ranged HI with great fidelity. The method relies on a new formulation for Ewald summation of the RPY tensor, which guarantees that the real-space and wave-space contributions to the tensor are independently symmetric and positive-definite for all possible particle configurations. Brownian displacements are drawn from a superposition of two independent samples: a wave-space (far-field) contribution, computed using techniques from fluctuating hydrodynamics and non-uniform fast Fourier transforms; and a real-space contribution, computed using a Krylov subspace method. The combined computational complexity of drawing these two independent samples scales linearly with the number of particles enabling hydrodynamic simulations with system sizes up to $10^6$ particles.

The short-ranged depletion attraction is modeled with an Asakura-Oosawa form[4]:

$$U_A(r) = U \frac{2(2a(1+\delta))^3 - 3r(2a(1+\delta))^2 + r^3}{2(2a(1+\delta))^3 - 6a(2a(1+\delta))^2 + (2a)^3}, \qquad (1)$$

for particle separations $r$ in the range of $2a < r < 2(a + \delta)$. The width of the attraction is $\delta = 0.057a$ and the value of the potential at contact, $U$, relative to the thermal energy scale, $kT$, is chosen to match the depletion strength of the polymer concentration $c/c^*$ used in the experiments.

Simulations of colloidal dispersions with 216,000 spherical particles of mean radius $a$ and 5% polydispersity are conducted in a cubic simulation box of size $L$ and periodic boundary conditions. We impose a particle volume fraction, $\phi = 0.20$, matching that of the experiments. The attractive dispersion is allowed to gel in a solvent of viscosity $\eta$ for $10^3$ bare diffusion steps $\tau_D = 6\pi\eta a^3/kT$ at which point each sample is found to be percolated and the measurements are performed. All

results are averaged over 5 independently generated samples for each value of $U/kT$.

In order to measure the structural correlation length in the simulated gels we compute the static structure factor using all particle positions, $\mathbf{x} \in \mathbb{R}^{3N}$:

$$S(q, t) = \left\langle \frac{1}{N} \sum_{j,k=1}^{N} \exp\left( i\mathbf{q} \cdot (\mathbf{x}_j(t) - \mathbf{x}_k(t)) \right) \right\rangle, \qquad (2)$$

$S(q, t)$ measures spatial correlations between particle positions over distances proportional to ~$2\pi/q$. As the samples are arrested, the $S(q, t)$ curves are unchanging at long times. The location of the peak in $S(q, t)$ at small wave vectors ($q \rightarrow 0$) is a measure of the characteristic correlation length of the particles in the gel. For each sample we identify the value of the wavevector at the maximum, $q_{max}$, using a local fit to a parabola, and use it to compute $\xi = 2\pi/q_{max}$.

## Data availability
The data that support the findings of this study are available from the corresponding authors upon reasonable request.

## Code availability
The simulation method, Brownian Dynamics with hydrodynamic interactions, is made publicly available through a plug-in to HOOMD-blue, the open source molecular dynamics suite, at the authors website: http://web.mit.edu/swangroup/software.shtml, as well as via email requests. HOOMD-blue version 1.3.4 is used in this study.

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

## Acknowledgements

E.M.F. and M.J.S. acknowledge funding from the International Fine Particles Research Institute. E.M.F. also thanks the National Science Foundation (CBET-1235955) and fruitful discussions with A. Lemaître. J.W.S. acknowledges ACS-PRF Grant No. 56719-DNI9 and an NSF CAREER grant (CBET-1554398) for financial support.

## Author contributions

E.M.F., M.J.S and J.W.S. designed the research; K.A.W. and L.C.H. collected the experimental data; K.A.W., L.C.H., E.M.F., M.J.S and J.W.S. analyzed data; Z.V. and J.W.S. performed computer simulation and contributed to theory; and E.M.F., J.W.S., K.A.W., L.C.H. and M.J.S. wrote the manuscript.

## Additional information

**Competing interests:** The authors declare no competing interests.

