## [Peer Review File · Nature Communications]

Reviewers' comments:

Reviewer #1 (Remarks to the Author):

The paper discusses the elasticity of colloidal gels and, as the title indicates, the authors aim at identifying its origin. They use a combination of experiments and numerical simulations to investigate the elastic modulus, and its relationship to the gel structure, as a function of the strength of the depletion interactions between the colloidal particles that compose the gel.

The paper is certainly interesting and the well conceived combination of experiments and computational work is clearly a great plus. I find the analysis developed here useful and insightful. Nevertheless, several points are not clear, raising concerns on the overall story and its suitability for Nature Communications.

1) How are the major claims of the paper novel? It seems to me, that the scenario proposed by the authors, in which the elasticity of the colloidal gel under study is due to the packing of locally glassy clusters has been already proposed and tested several times in the literature. The authors mention some of the related papers, e.g. ref. 12, 13, or 18.

What are they saying really new and/or different from previous studies? Aren't they mainly just confirming the scenario proposed (and tested) in those studies? If there is something more specifically new, it does not come across as such in the manuscript: it should be more explicitly spelled out and demonstrated in the paper.

As far as I understand, the main message is that the gelation is driven by the depletion interactions, which need to be strong enough, and therefore it is likely to happen in the two phase region. Locally the packing of the particles tends to be glassy (unless there is a stronger tendency to other types of packing for specific systems, I guess), but what counts for the structural arrest and the rigidity is also the larger scale structure that is rather the result of the combination of volume fraction, interaction strength and quenching protocol. In the closing part of the paper, the authors state that "this has been long suspected", but it seems to me that there is instead plenty of demonstrations in the literature. For what concerns the structural arrest, all of this has been already discussed, for example in ref. 11, 33, 34, 37 or in Kroy et al. PRL 2004 but also in many other papers. In the case considered here, this larger scale structure corresponds to clusters that are more and more densely packed as the strength of the depletion interactions between the particles increases. For the mechanical response, one can find the main outcomes already in the refs mentioned above (12, 13, 18) and in others. As far as I can understand from this manuscript, the main new contribution is the quantitative agreement between experiments and simulations for the specific experimental system, and the finding that the increase in the modulus with the interactions strength seems to be due not to an increase of the average cluster size but rather to an increase in the number of connections between clusters and in their number density. While these results are interesting and certainly deserve publication in some form, in the present form I don't see them meet the standards of this journal in terms of novelty and wide relevance (see also point 3)).

2) On the one hand, the authors emphasize the cluster packing (see title and various points in the manuscript), on the other they show that what is increasing with the interaction strength is not only the cluster density but also the inter cluster connections, suggesting that it is not just a packing problem. In addition, just before the Results section, the authors mention a similarity with the elasticity (and possibly) plasticity of polycrystalline solids, which emerges just from few weak bonds between clusters. How are these different aspects reconciled in the picture proposed by the authors? How do they know that only a few weak bonds are important?

3) While the scenario proposed here for the elasticity of colloidal gels (packing of locally glassy

clusters) is certainly possible and realized in certain conditions, like the ones relevant to the experiments and simulations performed here (plus see the refs mentioned above and others in the literature), the authors seem to suggest that this is the only possible mechanism. Since colloidal gels are non-ergodic materials, their properties and likely also the mechanism that drives the emergence of an elastic response depend on their history and the kinetics of the gelation transition. In this study, the authors have chosen one specific protocol to form the gel and their results are protocol specific, as far as I understand. Can the authors demonstrate that they are not? If the results are protocol specific, the claims of generality in the title and in various parts of the paper cannot be justified. And a narrower scope may make the suitability for this journal questionable.

4) An important ingredient to establish that the origin of the elasticity is in the cluster connections, seems to be the requirement that the clusters are rigid (i.e., they are the elastically active units, able to transmit stresses through their connections). It is not clear how the rigidity of the clusters is established in the simulations. Do they use a constraint counting criterion or do they measure the local rigidity of part of the gel structures? Are they relying on the fact that the local density is close to the one predicted for the glassy structural arrest?

5) One of the difficulties (and probably the main one) in establishing the origin of elasticity in gels is the fact that identifying the elastically connected domains is far from trivial. The authors here assume that the related length scale that appears in the Cauchy-Born modulus is well captured by the averaged square subgraph contact length $\langle r^2 \rangle$ and by the correlation length ξ . How can they support this assumption? If they have a good argument supporting this choice for the specific experimental conditions (volume fractions, interaction strengths, gelation protocol etc), how can they be sure that the same would hold in other conditions? Couldn't there be other length scales, not obviously detectable in the structure, that can control the elastically active domains? For other type of gels, for example, it is known that density fluctuations do not necessarily capture those lengthscales. It is also known that in complex self-assembled solid structures one can construct different length scales that scale differently with different control parameters (e.g. volume fractions, interaction strengths...). Can the authors better justify their choice and exclude that other length scales may be relevant?

6) Looking at Figure 1, the comparison between experimental and numerical images, correlation length from density fluctuations and bond stiffnesses are great. Can the authors understand the discrepancies in the elastic modulus, especially at low attraction strength? How was the modulus measured in the simulations? I could not find any information about this. I would expect that one way to exploit the simulations would be to make different type of measurements and show that they are consistent. Another information that would be useful to provide is the structure factor from which the correlation length is extracted.

Reviewer #2 (Remarks to the Author):

The major claim of this paper is that the Cauchy Born theory for the elastic modulus correctly predicts the growth of G' with the strength of the attractive interaction IF the rigid clusters and their connectivity is calculated by using a l -balanced graph partitioning scheme. The analysis presented in the paper is convincing. In addition, the deduction that the internal volume fraction in these clusters decreases with increasing attraction, and coincides with the extrapolated attractive glass line is intriguing and provides insight into the source of the mechanical strength of these gels.

The missing piece that troubles me is the lack of any argument relating the l -cluster decomposition to the rigidity of these clusters. As I understand, the l value is chosen by linking the radius of gyration of

the cluster to the correlation length of the gel. While this is certainly a way of identifying clusters, there is no argument given as to why these are the rigid clusters that should enter the Cauchy-Born theory. There is certainly a posteriori evidence that this decomposition has physical relevance (both from the success of the Cauchy-Born theory and the internal volume fraction matching the extrapolated glass-transition density), however, there is no independent verification that these clusters are indeed rigid. This community, including some of the present authors have used rigidity-percolation analysis to explore rigidity of gels. Can the authors look at the connectivity inside the I-clusters and at least do a Maxwell counting? In addition, can the increase in the number of elastically active bonds be related to increased number of self-stress states?

I am not suggesting that the authors perform a complete rigidity percolation analysis but some analysis of the I-clusters that can show that clusters identified this way are rigid would strengthen the conclusions. I will strongly support the publication of this paper in Nature Communications if this concern can be addressed.

Reviewer #4 (Remarks to the Author):

In this work, the authors use graph theory techniques to explore the origin elasticity in colloidal gels. They use both an experimental model system and numerically simulated gels, and show that experimental and simulated gels have quantitatively similar elastic moduli and microscopic correlation lengths. In the simulated gels they use a graph decomposition to identify clusters within the gel and show that the gel elasticity can be quantitatively connected to cluster-cluster connections. They further show that the volume fraction of particles within these clusters decreases with increasing attraction, appearing to extrapolate the mode coupling theory (MCT) attractive glass line to higher attractive strengths. In addition to

The results of this novel graph-theoretic approach are interesting and appear to open a new window to understand the structure and properties of colloidal gels. I feel this work will have a high impact within the gel and colloid science communities, and could potentially inspire a broad array of researchers working on particulate systems to consider applying and adapting such approaches. While I feel the novelty and potentially broad impact of this work makes it appropriate for publication in Nature Communications, I feel there are some issues that should be addressed prior to publication.

My primary concern is the blurring between computational and experimental results. This work emphasises the use of both experiments and simulations in the bold paragraph, and touts their ability to characterise both the macroscopic rheology and microstructure of their experimental model gels. However, it appears that the graph decomposition is only carried out on simulated gels (unless I am mistaken, however Fig. 2 seems to only refer to simulations). Was this ever attempted with the experimental gels? While I understand that in confocal microscopy particle contacts can be somewhat ambiguous, it seems like this could be done with some sort of reasonable threshold criteria. I was especially confused by the exp. / simulation divide with Fig. 3. The caption for Fig 3A only directly refers to simulation results, though the lines are computed from the experimental values of G' ? In Fig. 3B the caption is ambiguous, though it seems the data from this work corresponds to the experimental data? Why not include both experimental and simulation results on this phase diagram? These captions and figures should be modified to make it easier to identify and distinguish simulation and experimental results.

Smaller issues:

What is the origin of the the turnover in the variance of the number density at small distances that is seen in the simulations but not in experimental gels? Does this point to some sort of small structural difference, or is it simply a resolution artefact?

How sensitive are the graph decomposition results to the choice of l^* ? Since this is a somewhat arbitrary parameter, I feel that some analysis to show that the results don't change significantly as long as l^* is within a reasonable range.

I feel that more details are required in explaining how the hard sphere equation of state is used to compute $z_c(\phi_c)$. In ref. 18 they even explicitly state that this computation is somewhat controversial.

The circles with a dot in Fig. 3B are very difficult to distinguish from normal circles, especially when the dot is covered by another line. Please consider an alternative symbol.

Reviewer #1 (Remarks to the Author):

The paper discusses the elasticity of colloidal gels and, as the title indicates, the authors aim at identifying its origin. They use a combination of experiments and numerical simulations to investigate the elastic modulus, and its relationship to the gel structure, as a function of the strength of the depletion interactions between the colloidal particles that compose the gel.

The paper is certainly interesting and the well conceived combination of experiments and computational work is clearly a great plus. I find the analysis developed here useful and insightful. Nevertheless, several points are not clear, raising concerns on the overall story and its suitability for Nature Communications.

Reviewer comment

1) How are the major claims of the paper novel? It seems to me, that the scenario proposed by the authors, in which the elasticity of the colloidal gel under study is due to the packing of locally glassy clusters has been already proposed and tested several times in the literature. The authors mention some of the related papers, e.g. ref. 12, 13, or 18.

What are they saying really new and/or different from previous studies? Aren't they mainly just confirming the scenario proposed (and tested) in those studies? If there is something more specifically new, it does not come across as such in the manuscript: it should be more explicitly spelled out and demonstrated in the paper.

As far as I understand, the main message is that the gelation is driven by the depletion interactions, which need to be strong enough, and therefore it is likely to happen in the two phase region. Locally the packing of the particles tends to be glassy (unless there is a stronger tendency to other types of packing for specific systems, I guess), but what counts for the structural arrest and the rigidity is also the larger scale structure that is rather the result of the combination of volume fraction, interaction strength and quenching protocol. In the closing part of the paper, the authors state that "this has been long suspected", but it seems to me that there is instead plenty of demonstrations in the literature. For what concerns the structural arrest, all of this has been already discussed, for example in ref. 11, 33, 34, 37 or in Kroy et al. PRL 2004 but also in many other papers. In the case considered here, this larger scale structure corresponds to clusters that are more and more densely packed as the strength of the depletion interactions between the particles increases. For the mechanical response, one can find the main outcomes already in the refs mentioned above (12, 13, 18) and in others. As far as I can understand from this manuscript, the main new contribution is the quantitative agreement between experiments and simulations for the specific experimental system, and the finding that the increase in the modulus with the interactions strength seems to be due not to an increase of the average cluster size but rather to an increase in the number of connections between clusters and in their number density. While these results are interesting and certainly deserve publication in some form, in the present form I don't see them meet the standards of this journal in terms of novelty and wide relevance (see also point 3)).

Our response: As the reviewer points out (and which we discuss in the manuscript) the connection between gel structure and rheology are the subjects of many studies. We draw from these, but have advanced the understanding of gels in the work reported here. While the principal elements we combine have nearly all been postulated before, they have never been directly tested or confirmed. We do this through a combination of novel experiments, simulations, and graph theoretic approaches. The other reviewers found sufficient novelty in the approach. We believe readers will as well.

As the reviewer notes, references 12 (Ramakrishnan et al.) and 13 (Koumakis et al.) measure and report the rheology and microstructure of these systems. They postulate the role of clusters. In the case of Ramakrishnan et al., clusters are inferred from x-ray scattering measurements of the microstructure. This is necessary to resolve the discrepancy between the elasticity calculated by (essentially, although it is not described as such) Cauchy-Born theory based on *particle* contacts and the measured elasticity from

rheology. This work was incomplete, in the sense that a *direct* connection between the rheology and structure could not be made, but it was a tantalizing view and an important advance. In fact, we reinterpret the experimental their experimental data in terms of our new framework to get a predictive description for elasticity.

Our approach also addresses the reviewer's concern regarding the limits of our study. The reviewer states, "the main new contribution is the quantitative agreement... for the specific experimental system". But this is not the case. Our experimental system and the materials used by Ramakrishnan et al. differ in chemical composition and physical size (particle diameters and polymer radius of gyration) by an order of magnitude. The Ramakrishnan study used silica nanoparticles. We use polymer latex microspheres. This is one reason that our results are compelling: the physics applies to many systems, spanning a wide range of length scales and chemistries, just as is found in the range of applications where depletion gels occur.

One further comment: Reference 18 (Zaccone et al.) advanced the picture of cluster interactions, developing a stronger theoretical basis to interpret results such as those in the previously-cited experimental work. In our opinion, it is a beautiful and succinct picture of the (possible) mechanism of gel elasticity. However, as a work of theory, they could never directly draw this connection, such as by imaging the clusters, nor have an equation of state for the clusters. They explicitly say that they *imagine* this formulation only for high concentration dispersions (closer to an attractive glass than a more dilute gel). Ramakrishnan et al. similarly assume a cluster-like structure, inferring their existence from scattering data.

Reviewer comment

2) On the one hand, the authors emphasize the cluster packing (see title and various points in the manuscript), on the other they show that what is increasing with the interaction strength is not only the cluster density but also the inter cluster connections, suggesting that it is not just a packing problem. In addition, just before the Results section, the authors mention a similarity with the elasticity (and possibly) plasticity of polycrystalline solids, which emerges just from few weak bonds between clusters. How are these different aspects reconciled in the picture proposed by the authors? How do they know that only a few weak bonds are important?

Our response: We appreciate the author's confusion with respect to referencing polycrystalline solids. The intention was only to draw an analogy between the structure-property relations between gels and pioneering work for crystalline materials, the understanding of which revolutionized metallurgy in the 20th century and continues to provide important insights into the design of new nanomaterials. The suggestion is a good one, and we modified the manuscript to remove references (except at the end) to polycrystalline solids, since these references are ultimately a distraction early in the paper.

To the reviewer's point of the need to describe both cluster density and contacts, cluster density determines the cluster-cluster connections. They are not independent, but is a consequence of packing things differently. The work of Zaccone et al. describe this situation, asking as cluster number increases, how do and the resulting interactions affect the elasticity. Again, however, there was previously no direct way to count these contacts or directly observe the cluster packing. The graph theoretic approach here provides this ability.

Reviewer comment

3) While the scenario proposed here for the elasticity of colloidal gels (packing of locally glassy clusters) is certainly possible and realized in certain conditions, like the ones relevant to the experiments and simulations performed here (plus see the refs mentioned above and others in the literature), the authors seem to suggest that this is the only possible mechanism. Since colloidal gels are non-ergodic materials, their properties and likely also the mechanism that drives the emergence of an elastic response depend on

their history and the kinetics of the gelation transition. In this study, the authors have chosen one specific protocol to form the gel and their results are protocol specific, as far as I understand. Can the authors demonstrate that they are not? If the results are protocol specific, the claims of generality in the title and in various parts of the paper cannot be justified. And a narrower scope may make the suitability for this journal questionable.

Our response: The reviewer is very insightful. We don't disagree, and in fact highlight the possible "path-dependence" of gel formation in the discussion and conclusions. The ability to test our hypothesis (the validity of the Cauchy-Born theory of elasticity, the connectivity of glassy clusters) depends on a well-controlled and reproducible protocol in both experiment and simulation. Both the experiments and the simulations start from a well-dispersed state. This does not preclude other paths to other structures. Importantly, the theory and the analysis method are fundamentally tensorial, and should be able to capture not only differences in an isotropic microstructure, but anisotropies in the microstructure induced by processing. In future work, the graph decomposition method should enable investigators to examine those types of structures, and describe their mechanical properties. It is not a feature of this study, but certainly an important opportunity for future work.

We have revised the manuscript to include a discussion of the tensorial nature of the methods introduced in our work and these possibilities.

Reviewer comment

4) An important ingredient to establish that the origin of the elasticity is in the cluster connections, seems to be the requirement that the clusters are rigid (i.e., they are the elastically active units, able to transmit stresses through their connections). It is not clear how the rigidity of the clusters is established in the simulations. Do they use a constrain counting criterion or do they measure the local rigidity of part of the gel structures? Are they relying on the fact that the local density is close to the one predicted for the glassy structural arrest?

Our response: The question is similar to Reviewer 2's. In the simulations we note that the particles are constrained dynamically (the intermediate scattering function shows a plateau at long times) and the structure is elastic; these provide ample evidence of a frozen microstructure on all length scales. However our further analysis of the mean contact number in the clusters (described above) is consistent with a rigid, isostatic structure. We revised the text and included a new supplemental figure to address this point.

Reviewer comment

5) One of the difficulties (and probably the main one) in establishing the origin of elasticity in gels is the fact that identifying the elastically connected domains is far from trivial. The authors here assume that the related length scale that appears in the Cauchy-Born modulus is well captured by the averaged square subgraph contact length $\langle r^2 \rangle$ and by the correlation length ξ . How can they support this assumption? If they have a good argument supporting this choice for the specific experimental conditions (volume fractions, interaction strengths, gelation protocol etc), how can they be sure that the same would hold in other conditions? Couldn't there be other length scales, not obviously detectable in the structure, that can control the elastically active domains? For other type of gels, for example, it is known that density fluctuations do not necessarily capture those length scales. It is also known that in complex self-assembled solid structures one can construct different length scales that scale differently with different control parameters (e.g. volume fractions, interaction strengths...). Can the authors better justify their choice and exclude that other length scales may be relevant?

Our response: The reviewer asks how ξ , essentially the size of clusters, can be identified as the principle length scale determining the elasticity of the gel. This is, basically, our hypothesis (and the one gleaned from earlier theoretical work), which our experiments and simulations were designed to test. Do the

results hold under other conditions? We asked the same question, which motivated us to test our assumptions by examining the data of Ramakrishnan et al., which employ particles of different size and chemistry. Of course, the phenomena we describe and analyze do not preclude other types of particle gels and elastic behavior. Fractal gels are a good example. These gels form under conditions of much higher attractive strength and at much lower particle volume fractions. We took care to illustrate this distinction in the introduction and motivation for the work. For many years, fractal models have successfully described the rheology (elasticity, yield) of such colloidal gels; yet, such models fail for gels of moderate concentration and weaker attractive strength.

In terms of alternative length scales, we explicitly consider the “bond” and the corresponding “bond rigidity” as a potential key length scale, and perhaps the most readily identifiable. We exclude this length scale as it cannot account, even qualitatively, for the behavior of the elastic modulus with respect to attractive strength (polymer concentration.) It gives the wrong concavity of the modulus with respect to concentration and does not predict the overall magnitude of change in the elastic modulus at all.

Reviewer comment

6) Looking at Figure 1, the comparison between experimental and numerical images, correlation length from density fluctuations and bond stiffnesses are great. Can the authors understand the discrepancies in the elastic modulus, especially at low attraction strength? How was the modulus measured in the simulations? I could not find any information about this. I would expect that one way to exploit the simulations would be to make different type of measurements and show that they are consistent. Another information the would be useful to provide is the structure factor from which the correlation length is extracted.

Our response: We addressed a similar question raised by reviewer 4 and have included the full frequency-dependent moduli of both the experimental and simulated gels. They are in exceptional agreement. In terms of the simulation methods, used, we added a description to the supplemental information: “In the simulations, a swept-sine strain signal was used to deform the periodic box containing the colloidal gel in oscillatory simple shear. The shear stress was computed at each point in time, and the ratio of the Fourier transform of the stress and strain was used to define the complex modulus.”

Reviewer #2 (Remarks to the Author):

The major claim of this paper is that the Cauchy Born theory for the elastic modulus correctly predicts the growth of G' with the strength of the attractive interaction IF the rigid clusters and their connectivity is calculated by using a l -balanced graph partitioning scheme. The analysis presented in the paper is convincing. In addition, the deduction that the internal volume fraction in these clusters decreases with increasing attraction, and coincides with the extrapolated attractive glass line is intriguing and provides insight into the source of the mechanical strength of these gels.

The missing piece that troubles me is the lack of any argument relating the l -cluster decomposition to the rigidity of these clusters. As I understand, the l value is chosen by linking the radius of gyration of the cluster to the correlation length of the gel. While this is certainly a way of identifying clusters, there is no argument given as to why these are the rigid clusters that should enter the Cauchy-Born theory. There is certainly a posteriori evidence that this decomposition has physical relevance (both from the success of the Cauchy-Born theory and the internal volume fraction matching the extrapolated glass-transition density), however, there is no independent verification that these clusters are indeed rigid. This

community, including some of the present authors have used rigidity-percolation analysis to explore rigidity of gels. Can the authors look at the connectivity inside the l-clusters and at least do a Maxwell counting ? In addition, can the increase in the number of elastically active bonds be related to increased number of self-stress states?

I am not suggesting that the authors perform a complete rigidity percolation analysis but some analysis of the l-clusters that can show that clusters identified this way are rigid would strengthen the conclusions. I will strongly support the publication of this paper in Nature Communications if this concern can be addressed.

Our response: This is a great point: that we make no argument as to why clusters identified by the radius of gyration in the l-graph decomposition are the rigid clusters that should enter the Cauchy-Born theory. To address this, we performed the following analysis. Each cluster found back the decomposition is isolated from the gel and analyzed. The number of bonded neighbors is computed for each particle in each cluster and the distribution of bonded neighbors is computed. For all values of the attraction strength, U/kT , we find that mean number of bonds for particles within the clusters is 6 or higher, exceeding the requirement for isostaticity. Thus, the clusters reflect an elastic core connected to other clusters through the few bonds cut during the decomposition. We revised the manuscript by including this point and provide a new supplemental figure of the particle nearest neighbor analysis.

New Supplemental Figure 3: The nearest neighbor distribution for particles within clusters identified by the l-balanced graph decomposition.

Reviewer #4 (Remarks to the Author):

In this work, the authors use graph theory techniques to explore the origin elasticity in colloidal gels. They use both an experimental model system and numerically simulated gels, and show that experimental and simulated gels have quantitatively similar elastic moduli and microscopic correlation lengths. In the simulated gels they use a graph decomposition to identify clusters within the gel and show that the gel

elasticity can be quantitatively connected to cluster-cluster connections. They further show that the volume fraction of particles within these clusters decreases with increasing attraction, appearing to extrapolate the mode coupling theory (MCT) attractive glass line to higher attractive strengths. In addition to

The results of this novel graph-theoretic approach are interesting and appear to open a new window to understand the structure and properties of colloidal gels. I feel this work will have a high impact within the gel and colloid science communities, and could potentially inspire a broad array of researchers working on particulate systems to consider applying and adapting such approaches. While I feel the novelty and potentially broad impact of this work makes it appropriate for publication in Nature Communications, I feel there are some issues that should be addressed prior to publication.

Reviewer comment

My primary concern is the blurring between computational and experimental results. This work emphasises the use of both experiments and simulations in the bold paragraph, and touts their ability to characterise both the macroscopic rheology and microstructure of their experimental model gels. However, it appears that the graph decomposition is only carried out on simulated gels (unless I am mistaken, however Fig. 2 seems to only refer to simulations). Was this ever attempted with the experimental gels? While I understand that in confocal microscopy particle contacts can be somewhat ambiguous, it seems like this could be done with some sort of reasonable threshold criteria.

Our response: The reviewer raises an excellent point - it is certainly desirable to apply the graph theory to both experiment and simulation. As noted, there is some discrepancy in particle positions detected in confocal images, a well-known limit of the experiment. These generally contribute minimally to the results reported previously in the literature - e.g. measures of structure, both local (bond number, radial distribution function) and global (number density fluctuations). However, the present experimental system sacrifices some resolution (index mismatch) to enable both rheology and optical trapping, as described in our earlier work. This increases both the static and dynamic contributions to the variation in particle positions. The simulations nicely circumvent this limitation and make the graph theoretic approach possible. Figure 1D clearly shows that small wave vectors and short length scales of these contributions, while both experiment and theory capture the long-range structure of the gel. It is clear from our work that there is an exciting opportunity to apply the graph-theoretic approach directly to experimental systems as the time- and spatial-resolution of imaging systems improves.

The key issue here is to establish, with as much certainty as possible, the agreement between the simulations and experimental structure and rheology. Our submitted manuscript only compared the elastic modulus at low frequency - corresponding to the predicted values of Cauchy-Born theory. We performed a comparison of both the storage and loss modulus over the entire range of measured frequencies, which confirm a quantitative agreement between simulation and experiment. The correct frequency dependence confirms that the structure should be the same on the smaller length scales in question. We include this as a supplemental figure in our revised manuscript, which is also included here:

New Supplemental figure 2: The linear viscoelasticity for the six different gels investigated experimentally and in simulations. Closed symbols are experimental measurements on a strain controlled rheometer. Open symbols are the results of dynamic simulations. The polymer concentrations correspond to symbol colors as follows $c/c^* = 0.35$ (light blue), 0.47 (magenta), 0.59 (green), 0.71 (dark blue), 0.83 (red), 0.94 (black).

Reviewer comment

I was especially confused by the exp. / simulation divide with Fig. 3. The caption for Fig 3A only directly refers to simulation results, though the lines are computed from the experimental values of G' ? In Fig. 3B the caption is ambiguous, though it seems the data from this work corresponds to the experimental data? Why not include both experimental and simulation results on this phase diagram? These captions and figures should be modified to make it easier to identify and distinguish simulation and experimental results.

Our response: This is a great suggestion, and we have added the simulation results to figure 3B. We updated the caption to eliminate the ambiguity.

Reviewer comment

Smaller issues:

What is the origin of the the turnover in the variance of the number density at small distances that is seen in the simulations but not in experimental gels? Does this point to some sort of small structural difference, or is it simply a resolution artefact?

Our response: This turnover is explained above. It is an artifact due to the particle tracking resolution.

Reviewer comment

How sensitive are the graph decomposition results to the choice of l^ ? Since this is a somewhat arbitrary parameter, I feel that some analysis to show that the results don't change significantly as long as l^* is within a reasonable range.*

Our response: l^* is a result of the graph decomposition correlated with the microstructure and not an input parameter. The author is asking if the n_c , z_c and $\langle r^2 \rangle$ vary significantly in the neighborhood of a particular value of l^* . They do not, and we know this because the bisection algorithm used to find l^* is hunting for the local minimum of an objective function that was found to vary smoothly and slowly in the neighborhood of l^* (and for all l much smaller than number of particles in the gel). We attribute this smoothness to the size of the simulated gels relative to the size of the rigid clusters in the gel. $O(100$'s) of clusters are identified with the graph decomposition. Another way to ask this question is to repeat the analysis with different initial conditions for the clustering algorithm, which we did in preparation of figure 2. The error bars (uncertainties on n_c , z_c and $\langle r^2 \rangle$) are smaller than the symbols in the figure. We have commented on this in the revised text.

Reviewer comment

I feel that more details are required in explaining how the hard sphere equation of state is used to compute $z_c(\phi_c)$. In ref. 18 they even explicitly state that this computation is somewhat controversial.

Our response: The controversy lies in whether clusters pack as hard spheres. This is clearly a necessary simplifying assumption in previous work. Here we establish the quality of the approximation in Figure 3A by comparing the values to those derived from graph theory. The values agree within 10%. We revised the manuscript to clarify that at each experimental condition, we solved the implicit equation for ϕ_c . We now also include the equations for the contact number.

Reviewer comment

The circles with a dot in Fig. 3B are very difficult to distinguish from normal circles, especially when the dot is covered by another line. Please consider an alternative symbol.

Our response: We thank the reviewer for the suggestion that makes this figure clearer. The symbols have been updated in the revised manuscript.

Reviewers' comments:

Reviewer #1 (Remarks to the Author):

While I appreciate some of the revisions made, I find that the revised manuscript fails at clarifying the main novelty of the paper. I note also that the authors have not properly addressed a few important points I made in my report. I invite them therefore to revise further the manuscript, to make sure the following points are taken care of.

1) The main outcome of this paper (as the title states: "Colloidal Gel Elasticity Arises From the Packing of Locally Glassy Clusters") is an independent demonstration, through experiments and simulations, that the picture and the theory proposed in Ref. 16 work.

Ref. 16 proposes that the elasticity in dense colloidal gels results from the glassy structural arrest of clusters formed during colloidal aggregation. Contrarily to what the authors suggest in the manuscript and stated in the answer to my previous report, Ref. 16 does draw a direct connection to experiments. Figs. 1 and 2 of that paper contain a comparison between the theory and the experiments from Ref. 10 and from Laurati et al., J Chem Phys 2009 (also relevant to this work). The comparison clearly shows that the modulus measured in the experiments cannot be ascribed to particle-particle connectivity in the dense phase (the attractive glass) but can be instead understood in terms of cluster-cluster connections, if one uses the estimate provided from the experiments for the cluster size and the particle-particle interactions.

In the present version of this manuscript, the authors seem to suggest at several points that a new picture and a different, independent theory have been developed here, which is misleading. Different from the previous experiments, here the authors are able to identify the clusters in real space and use the numerical simulations to obtain a more accurate estimate of the average number density of the clusters n_c , the average number of contacts z_c , and an average length scale that is a proxy of the correlation length measured through density fluctuations. Through this analysis the authors find that, while the length scale estimated depends weakly on the interaction strength, the combination $n_c z_c$ is instead a convex function of the interaction strength, as it is the modulus measured in the rheology. This information is contrasted by the dependence of the inter-particle bond stiffness on the interaction strength (as computed from AO theory) which shows rather a concave behavior.

By using the length-scale, n_c and z_c from numerical data in the theory proposed in Ref. 16, the authors obtain an estimate of the modulus that is in good agreement with the experiments. Further, the authors estimate the volume fraction of the clusters and of the particles inside the clusters using the hard sphere equation of state in the theory of Ref. 16 and comment how the values obtained can be seen as an extension of the glass transition lines in sticky colloids calculated in other works.

Hence, the main novel result is that the picture and the theory of Ref. 16 works also for these independently conceived experiments/simulations and that the modulus of the gels can indeed be explained in terms of cluster contributions to the elasticity, as suggested there.

In Ref. 16 it was also proposed, for simplicity and because inspired by previous work (such as Kroy et al., PRL 2004 and Del Gado et al. PRE 2004) that the clusters behave approximately as glassy hard spheres. Hence they used a hard sphere EOS to estimate their average contact number (when not known) from the particle volume fractions.

Such assumption was proven reasonable by the comparison with the experiments mentioned above. (As a side remark, any major difference from hard spheres predictions would be actually surprising, in view of the limited range for volume fractions and contact numbers close to structural arrest.)

Hence it is not surprising, but reassuring, that using the hard sphere equation of state here (where instead the average contact number is known), yield reasonable numbers for the cluster and the particle volume fractions. The estimates of those volume fractions sort of line up with the glass lines in attractive colloidal suspensions, in agreement with previous works stating that the dense phase of colloidal gels is akin to an attractive glass (see also P. Lu et al, Nature 2008, but not only) and again in agreement with Ref. 16 proposition that the elasticity emerges from a glassy assembly of clusters.

To summarize, there is no new insight gained from this paper that was not already proposed in Ref. 16. An independent experimental proof of a previously existing theoretical picture, through a combination of experiments and simulations together with the graph analysis developed for the numerical data, is the way to drive progress. It is valuable and can be suitable for Nature Communication, if the contribution of Ref. 16 is honestly acknowledged.

Suggesting, as the text does in several points, that Ref. 16 merely provided a formula (or a Cauchy Born theory) to compute the modulus is misleading.

I also note that, in this paper, the combination of experiments/simulations and the graph analysis alone, on the other hand, would not justify the novelty and broad impact required for this journal:

a) The direct imaging of the clusters is a nice feature of this work. Nevertheless, this is certainly not the first time that clusters are imaged in colloidal gels (see, in addition to Lu et al. Nature 2008; Dinsmore and Weitz, JPCM 2002; Hsiao et al, PNAS 2008; Royall et al, Nature Mat. 2008; and many more). Moreover I cannot share the view (stated by the authors in the answer to my previous report and also suggested in the paper) that imaging provides the true direct evidence that is missing in scattering techniques, in this context. Scattering, when used correctly and rigorously, does provide quantitative structural characterization and for sure I don't need to point out the sheer amount of scientific insight gained through scattering techniques in colloidal systems because of that. The scattering studies mentioned here and others (see Ref. 10, but not only) do provide quantitative structural information about the clusters.

Combining confocal experiments with simulations that show very similar structures for colloidal gels has also been achieved in other works (see again P. Lu et al. Nature 2008; Koumakis et al., Soft Matter 2015; E. Sanz et al., PRL 2009; Hsiao et al. PNAS 2008).

b) The graph analysis is interesting and valuable as well, but, as the authors have clarified, it is limited to the simulations, since applying it to the experiments is still not possible. If restricted to the simulations, there are several methods to analyze connectivity in condensed matter physics that could be equally or more valuable. Hence, again, the case for broad impact and novelty is arguable, since the method is not unique and its performance is not directly compared to others.

c) The authors have made a further point about the cluster rigidity in the revised manuscript and in the answer to the referees. While I appreciate this point and I think it is valuable, one would need a proper rigidity analysis for it to be the prerogative of this manuscript. Counting the number of contacts is a necessary but non-sufficient condition and in fact more sophisticated methods have been devised to determine the rigidity when one cannot measure it directly (see for example the pebble game). On the other hand, similar analysis of the contact number distributions in colloidal gels have been provided in several papers.

Here there is no other evidence provided, beyond the local coordination number, that the clusters are rigid. Referee 2 and myself were in fact asking for data to support the possibility that the clusters are rigid and the data on the coordination number go in the right direction. Stating, only on the basis of

those data, that the authors prove here that the clusters are rigid is misleading.

Recent papers that the authors may want to mention to support the importance of rigidity in colloidal gels are: ArXiv.1804.04370 and ArXiv.1807.08858.

More specific points in the text that require attention:

- Final discussion: the sentence "These results confirm what has long been suspected - that there is an intimate connection between the gelation of colloids with short-range attraction and phase separation" sounds really naive and inappropriate. More than 20 years of literature on colloidal gelation cannot be called "a suspicion" (the first paper by Carpineti & Giglio about spinodal decomposition in colloidal gels is from 1994). This include of course even a lot of the papers the authors themselves cite.

- Strictly speaking, finding a value of the volume fraction that is close to a glass transition value does not guarantee that a system is a glass, since the glassy state depends on the protocol and needs to be characterized through the dynamics. Again, this is a necessary but not sufficient condition.

Reviewer #2 (Remarks to the Author):

I have read the revised manuscript and the response to referees. The authors have addressed my concerns satisfactorily, and I am happy to support publication.

Reviewer #4 (Remarks to the Author):

In their revised manuscript, the authors have satisfactorily addressed all the issues raised in my initial review. The added content in the SI, particularly the analysis of the number of bonded neighbors within the cluster added in response to reviewers #1 and #2, strengthens the papers conclusions.

Reviewer #1 raised the issue of the novelty of this work, particularly in light of previous work suggesting a link between gel elasticity and local clusters. It is my opinion that the network analysis performed here goes beyond this previous previous to quantitatively relate the gel microstructure and rheology in a novel fashion. I affirm my initial assessment that this work represents a significant advance and, now the my initial comments have been addressed, I strongly recommend publication in Nature Communications.

Reviewer #1:

Reviewer comment:

To summarize, there is no new insight gained from this paper that was not already proposed in Ref. 16. An independent experimental proof of a previously existing theoretical picture, through a combination of experiments and simulations together with the graph analysis developed for the numerical data, is the way to drive progress. It is valuable and can be suitable for Nature Communication, if the contribution of Ref. 16 is honestly acknowledged.

Our response:

This seems to be the core issue for the reviewer and we have sought to rectify it. Hence, this comment has been elevated to the top of the response despite appearing much later in the reviewer's original comments. A copy of this response has been left in place in order to preserve the record.

We have added the following to the text when [16] is first cited:

“Linear elasticity in depletion gels has been postulated to result from the spatial organization of particles into locally dense clusters, each cluster acting as a rigid, mechanical unit that propagates the elastic deformation (10, 15, 16). For instance, the work of Zaccone, Wu, and Del Gado (16) identified the role of cluster-cluster contacts in gel elasticity. The authors showed that one model of cluster microstructure, based on the contact number distribution for hard spheres, when combined with the Cauchy-Born theory of elasticity in amorphous solids, can fit experimental measurements of the shear modulus. Similar models with different approaches to enumerating clusters and cluster contacts based on mode coupling theory were also explored by Ramakrishnan and co-workers (10).

...

In the present work, we combine experimental, computational, and graph-theoretic approaches to systematically identify clusters in depletion gels and show that they indeed constitute the fundamental structural units that lead to the gel's elastic response.”

This extra step properly acknowledges the role of [16] in bringing the Cauchy-Born theory to the space of colloidal gelation and postulating some of the microstructural details. We believe these modifications satisfy the conditions for publication presented by the reviewer.

Reviewer comment:

1) The main outcome of this paper (as the title states: "Colloidal Gel Elasticity Arises From the Packing of Locally Glassy Clusters") is an independent demonstration, through experiments and simulations, that the picture and the theory proposed in Ref. 16 work.

Ref. 16 proposes that the elasticity in dense colloidal gels results from the glassy structural arrest of clusters formed during colloidal aggregation. Contrarily to what the

authors suggest in the manuscript and stated in the answer to my previous report, Ref. 16 does draw a direct connection to experiments. Figs. 1 and 2 of that paper contain a comparison between the theory and the experiments from Ref. 10 and from Laurati et al., J Chem Phys 2009 (also relevant to this work). The comparison clearly shows that the modulus measured in the experiments cannot be ascribed to particle-particle connectivity in the dense phase (the attractive glass) but can be instead understood in terms of cluster-cluster connections, if one uses the estimate provided from the experiments for the cluster size and the particle-particle interactions.

Our response:

We have done more than independently demonstrate that the theory in [16] is a good descriptor of colloidal gel elasticity. The theory in [16] needs models to quantify the stiffness of inter-cluster contacts, the pair distribution function of clusters, and the cluster size. In making a comparison with the experimental data from the work by Laurati, the authors of [16] estimate these quantities via sensible empiricisms available at the time. For example, from [16]: “Again for densely packed clusters (i.e., with volume fraction in the range 0.58–0.64) we estimate $\phi_c \approx 0.6$ (and check that variation around this value does not significantly change our outcomes).”

It should be evident from our manuscript that we were in part inspired to perform our study by the theory advocated in [16]. We are not the progenitors of the theory, we do not claim to be. We have sought in this work to connect the Cauchy-Born theory, which was known before [16] for other amorphous materials, to the microstructural and micromechanical details of well-controlled colloidal gels. We have gained insight about gels from the development of an unbiased procedure to parameterize the model. Figures 1 and 2 in [16] are incomplete, in our opinions, and could be improved on. The analysis in [16], for example, *cannot* explain the concavity of the elastic modulus measured in this work with respect to the interparticle attractive strength. Yet, this behavior would be central to the design or prediction of a weak colloidal gel’s elasticity. The method of analysis advocated in this work can explain the concavity observed both in experiments cited in [16] and in our own experiments.

Reviewer comment:

In the present version of this manuscript, the authors seem to suggest at several points that a new picture and a different, independent theory have been developed here, which is misleading. Different from the previous experiments, here the authors are able to identify the clusters in real space and use the numerical simulations to obtain a more accurate estimate of the average number density of the clusters n_c , the average number of contacts z_c , and an average length scale that is a proxy of the correlation length measured through density fluctuations. Through this analysis the authors find that, while the length scale estimated depends weakly on the interaction strength, the combination $n_c z_c$ is instead a convex function of the interaction strength, as it is the modulus measured in the rheology. This information is contrasted by the dependence of

the inter-particle bond stiffness on the interaction strength (as computed from AO theory) which shows rather a concave behavior. By using the length-scale, n_c and z_c from numerical data in the theory proposed in Ref. 16, the authors obtain an estimate of the modulus that is in good agreement with the experiments. Further, the authors estimate the volume fraction of the clusters and of the particles inside the clusters using the hard sphere equation of state in the theory of Ref. 16 and comment how the values obtained can be seen as an extension of the glass transition lines in sticky colloids calculated in other works.

Our response:

We *are not* misleading the reader. Our contribution is stated clearly in the early paragraphs of the manuscript, in which the theory of [16] is called out as the primitive on which the study is based:

“Linear elasticity in depletion gels has been postulated to result from the spatial organization of particles into locally dense clusters, each cluster acting as a rigid, mechanical unit that propagates the elastic deformation (10, 15, 16). Signatures of clustering are evident in light scattering (10) and confocal microscopy (17), which probe long-range fluctuations in colloid number density, and active microrheology, which examines the elastic deformation in response to a local perturbation (18). In the present work, we combine experimental, computational, and graph-theoretic approaches to systematically identify the fundamental structural units of depletion gels and predict the elastic response arising from these units. “

“To resolve this important puzzle, we use a graph theoretic approach to identify the fundamental elastic units of the gels, a set of clusters, and to measure the density of cluster-cluster contacts. The physical size of the clusters is independent of the strength of the depletion attraction, but their number density and the number of cluster-cluster contacts grows with increased attractive strength. This response, when combined with the Cauchy-Born theory for the affine elastic response of amorphous solids (16, 22), yields a prediction of the elastic modulus with the correct convexity and in quantitative agreement with experimental measurements and calculations from simulations.”

The theory along with the appropriate citations are also discussed at length. We neither claim to be the first to derive it nor the first to try to apply it to colloidal gels. We agree with the reviewer that it is important to give credit to past researchers and distinguish our present effort from those we cite. We have made additional effort to do so in the preceding responses.

Reviewer comment:

Hence, the main novel result is that the picture and the theory of Ref. 16 works also for these independently conceived experiments/simulations and that the modulus of the gels can indeed be explained in terms of cluster contributions to the elasticity, as suggested there.

In Ref. 16 it was also proposed, for simplicity and because inspired by previous work (such as Kroy et al., PRL 2004 and Del Gado et al. PRE 2004) that the clusters behave approximately as glassy hard spheres. Hence they used a hard sphere EOS to estimate their average contact number (when not known) from the particle volume fractions. Such assumption was proven reasonable by the comparison with the experiments mentioned above. (As a side remark, any major difference from hard spheres predictions would be actually surprising, in view of the limited range for volume fractions and contact numbers close to structural arrest.)

Our response:

Again, the reviewer is exaggerating the contributions of the work in [16], which are considerable, while underrating the scientific efforts in the present study. The authors of [16] speculate: that there are clusters at all, that the clusters are relatively homogeneous in nature, and that the number of cluster-cluster contacts can be described by the hard sphere equation of state. They do not know any of this to be the case, and for the other colloidal gels cited by the reviewer in their previous review, some of these postulates are almost certainly wrong. The authors of [16] also hypothesize that this picture should only hold at high volume fraction and look at cases with volume fraction greater than 40%.

In the present work, we have used a combination of methods to infer the underlying set of clusters and cluster contacts *directly*. That the results are in reasonable agreement with the postulates of [16] is not a demerit and should not be treated as such. With the graph analysis, we can apply the Cauchy-Born theory *without need for making assumptions about the microstructure*. The present results are for gels that are significantly less concentrated than in [16], and yet the Cauchy-Born theory still appears to work. This meets, in one aspect, the reviewer's definition of "surprising."

In addition to being able to connect microstructural and micromechanical features of the gel to the macroscopic elasticity *directly* through imaging and the graph theory, we are able to use the analysis to gain further insight into the nature of the clusters making up the network. The approach advocated for in this work is tested against a relatively homogenous and isotropic colloidal gel for which we find that a hard sphere like EOS is comparable to what the cluster analysis yields and the heterogeneity of the clusters is minimal. However, our approach can be extended to circumstances well beyond, and this is discussed in previous manuscript modifications suggested by this reviewer and others.

The other reviewers disagree with the above opinion on what should be considered novel or the value of various novelties. They each highlight how the careful experimentation, direct measurement, simulation, and the application of graph theoretic concepts leads to a significant advance over the previous state of the art.

Reviewer comment:

Hence it is not surprising, but reassuring, that using the hard sphere equation of state here (where instead the average contact number is known), yield reasonable numbers for the cluster and the particle volume fractions. The estimates of those volume fractions sort of line up with the glass lines in attractive colloidal suspensions, in agreement with previous works stating that the dense phase of colloidal gels is akin to an attractive glass (see also P. Lu et al, Nature 2008, but not only) and again in agreement with Ref. 16 proposition that the elasticity emerges from a glassy assembly of clusters.

Our response:

The volume fractions of the glassy clusters directly measured in this work are not the same as those measured by Lu. We include Lu's data in Fig. 4 of the paper and discuss it extensively. Distinct from [16] and the work of Lu is our claim that the clusters themselves follow the attractive glass line. [16] discusses "glassy arrest" of clusters and pays little mind to the internal structure of the clusters. What it means for clusters to undergo glassy arrest is also not discussed in any detail in [16]. At any rate, that the reviewer is not surprised that a proposition from [16] (which is not one we make in the present work) is born out through combined experiment, simulation, and new analytical methods is not a strike against the present work.

Reviewer comment:

To summarize, there is no new insight gained from this paper that was not already proposed in Ref. 16. An independent experimental proof of a previously existing theoretical picture, through a combination of experiments and simulations together with the graph analysis developed for the numerical data, is the way to drive progress. It is valuable and can be suitable for Nature Communication, if the contribution of Ref. 16 is honestly acknowledged.

Our response:

This seems to be the core issue for the reviewer and we have sought to rectify it. Hence, this comment has been elevated to the top of the response despite appearing much later in the reviewer's original comments. A copy of this response has been left in place in order to preserve the record. Specific changes are documented above.

Reviewer comment:

Suggesting, as the text does in several points, that Ref. 16 merely provided a formula (or a Cauchy Born theory) to compute the modulus is misleading.

I also note that, in this paper, the combination of experiments/simulations and the graph analysis alone, on the other hand, would not justify the novelty and broad impact required for this journal:

Our response:

Again, we have not mislead. This is addressed in prior responses. Whether this work meets the reviewer's standard for publication appears to hinge on how [16] is acknowledged within the text, and this is also addressed in prior responses and through our manuscript revisions.

Reviewer comment:

a) The direct imaging of the clusters is a nice feature of this work. Nevertheless, this is certainly not the first time that clusters are imaged in colloidal gels (see, in addition to Lu et al. Nature 2008; Dinsmore and Weitz, JPCM 2002; Hsiao et al, PNAS 2008; Royall et al, Nature Mat. 2008; and many more). Moreover I cannot share the view (stated by the authors in the answer to my previous report and also suggested in the paper) that imaging provides the true direct evidence that is missing in scattering techniques, in this context. Scattering, when used correctly and rigorously, does provide quantitative structural characterization and for sure I don't need to point out the sheer amount of scientific insight gained through scattering techniques in colloidal systems because of that. The scattering studies mentioned here and others (see Ref. 10, but not only) do provide quantitative structural information about the clusters.

Combining confocal experiments with simulations that show very similar structures for colloidal gels has also been achieved in other works (see again P. Lu et al. Nature 2008; Koumakis et al., Soft Matter 2015; E. Sanz et al., PRL 2009; Hsiao et al. PNAS 2008).

Our response:

We do not claim to be the first to do direct imaging of colloidal gels, but one of the co-authors is among the first to do exactly this. Again, we cite the appropriate literature including the early work of Dibble and Solomon.

We agree scattering techniques are great! There is no light source with high enough flux, fast enough sampling, and high enough resolution to enable the sort of analysis we do in this work. One cannot, as of yet, solve the structure of a colloidal gel via scattering as one would do for a biomacromolecule. Yet, this is the level of information required to do any analysis of connectivity.

Reviewer comment:

b) The graph analysis is interesting and valuable as well, but, as the authors have clarified, it is limited to the simulations, since applying it to the experiments is still not possible. If restricted to the simulations, there are several methods to analyze connectivity in condensed matter physics that could be equally or more valuable. Hence, again, the case for broad impact and novelty is arguable, since the method is not unique and its performance is not directly compared to others.

Our response:

We have not claimed that the graph analysis is limited to the simulations, only that the confocal imaging in the present study is not high enough resolution to allow reliable application to these gels. Citation 28 is a footnote describing limitations of the confocal microscopy experiment performed in this work and is discussed at great length in reference 19. As the reviewer notes, there are many methods of analyzing connectivity. “Better” is a subjective measure that depends on the circumstances. We considered other measures of connectivity. None was more informative (or informative at all) than the graph analysis for the present purpose of estimating the elastic modulus. In fact, as far as we are aware, the connectivity analysis we present is the only one to bear on this question. Finally, we explain in detail in the text how non-unique solutions are expressed in the graph analysis and that this has no bearing on the mechanics inferred from the analysis. Appealing to degeneracy in the set of sub-graphs unfairly devalues the methodology.

Reviewer comment:

c) The authors have made a further point about the cluster rigidity in the revised manuscript and in the answer to the reviewers. While I appreciate this point and I think it is valuable, one would need a proper rigidity analysis for it to be the prerogative of this manuscript. Counting the number of contacts is a necessary but non-sufficient condition and in fact more sophisticated methods have been devised to determine the rigidity when one cannot measure it directly (see for example the pebble game). On the other hand, similar analysis of the contact number distributions in colloidal gels have been provided in several papers.

Here there is no other evidence provided, beyond the local coordination number, that the clusters are rigid. Reviewer 2 and myself were in fact asking for data to support the possibility that the clusters are rigid and the data on the coordination number go in the right direction. Stating, only on the basis of those data, that the authors prove here that the clusters are rigid is misleading.

Recent papers that the authors may want to mention to support the importance of rigidity in colloidal gels are: ArXiv.1804.04370 and ArXiv.1807.08858.

Our response:

Reviewer 2 specifically asked for the contact number distribution and we provided it. The proposition of [16] and this work is that the clusters are much stiffer than the network as a whole. All the evidence we have collected supports this claim. We state

that: “For particles with central interactions, six bonds are necessary for iso-staticity and thus rigidity,” which is correct. We modify the next sentence to read: “The contact number distribution cannot prove definitively that the clusters identified by the decomposition are internally rigid. However, that the mean number of contacts per particle in a cluster is larger than six is consistent with clusters having a rigid core.”

Regarding the ArXiv papers. These are very nice works and we have added citations emphasizing the role of rigidity in gelation that they study.

Reviewer comment:

More specific points in the text that require attention:

- Final discussion: the sentence "These results confirm what has long been suspected - that there is an intimate connection between the gelation of colloids with short-range attraction and phase separation" sounds really naive and inappropriate. More than 20 years of literature on colloidal gelation cannot be called "a suspicion" (the first paper by Carpineti & Giglio about spinodal decomposition in colloidal gels is from 1994). This include of course even a lot of the papers the authors themselves cite.

- Strictly speaking, finding a value of the volume fraction that is close to a glass transition value does not guarantee that a system is a glass, since the glassy state depends on the protocol and needs to be characterized through the dynamics. Again, this is a necessary but not sufficient condition.

Our response:

We have edited the final discussion to read: “These results are supported by a long line of evidence that there is an intimate connection between the gelation of colloids with short-range attraction and phase separation. For particles with centro-symmetric forces, a thermodynamic driving force in the two-phase region drives aggregation, and locally arrested density fluctuations, clusters that are sparsely connected, give the network elastic properties.”

With regard to the second comment, we have revised the discussion regarding volume fraction of the attractive glass to highlight current and future studies following the dynamics of clusters identified via the graph decomposition. Our revised sentence reads: “It is important to note that glassiness is conventionally defined in terms of long or diverging relaxation times and not merely the density of a phase. We observe in both experiments and simulations that the gel as a whole is arrested. This highlights the importance of work that tracks the morphology and dynamics of individual clusters over longer time scales than studied here, in order to understand how clusters form, relax, and eventually arrest (31).”

Reviewer #2:

Reviewer comment:

I have read the revised manuscript and the response to reviewers. The authors have addressed my concerns satisfactorily, and I am happy to support publication.

Our response:

We thank the reviewer for their helpful comments.

Reviewer #4:

Reviewer comment:

In their revised manuscript, the authors have satisfactorily addressed all the issues raised in my initial review. The added content in the SI, particularly the analysis of the number of bonded neighbors within the cluster added in response to reviewers #1 and #2, strengthens the papers conclusions.

Reviewer #1 raised the issue of the novelty of this work, particularly in light of previous work suggesting a link between gel elasticity and local clusters. It is my opinion that the network analysis performed here goes beyond this previous previous to quantitatively relate the gel microstructure and rheology in a novel fashion. I affirm my initial assessment that this work represents a significant advance and, now the my initial comments have been addressed, I strongly recommend publication in Nature Communications.

Our response:

We thank the reviewer for their helpful comments and support.

REVIEWERS' COMMENTS:

Reviewer #1 (Remarks to the Author):

I appreciate the authors' attention to my comments and the new modifications they have made to the manuscript.

By no means I consider the agreement between the findings presented here and the picture proposed in ref.[16] a point against this work. The criticisms were not aimed at underrating the work done here, which I value.

My comments were instead aimed at correcting what I considered misleading statements in the previous version of the manuscript and in the answer to my first report.

The authors have now addressed the points I raised and modified the manuscript accordingly. The new revised version of the manuscript more accurately describes the contribution of this work.

On this basis, I can recommend publication.

Minor points:

- Laurati et al. J. Chem. Phys. 2009 could be included in the references.
- page 11, the authors refer to ref. 16 by mentioning only 2 (Zaccone and Del Gado) of the 3 authors (Zaccone, Wu and Del Gado).

Reviewer #1:

Reviewer comment:

I appreciate the authors' attention to my comments and the new modifications they have made to the manuscript.

By no means I consider the agreement between the findings presented here and the picture proposed in ref.[16] a point against this work. The criticisms were not aimed at underrating the work done here, which I value.

My comments were instead aimed at correcting what I considered misleading statements in the previous version of the manuscript and in the answer to my first report.

The authors have now addressed the points I raised and modified the manuscript accordingly. The new revised version of the manuscript more accurately describes the contribution of this work.

On this basis, I can recommend publication.

Minor points:

- Laurati et al. J. Chem. Phys. 2009 could be included in the references.*
- page 11, the authors refer to ref. 16 by mentioning only 2 (Zaccone and Del Gado) of the 3 authors (Zaccone, Wu and Del Gado).*

Our response:

We thank the reviewer for their helpful comments throughout the review process. The minor points raised by the reviewer have been addressed. We added a citation to the gel rheology studies of Laurati et al. in the introduction. We have also corrected our oversight in referring to the work of Zaccone, Wu, and Del Gado and thank the reviewer for pointing out our error.